# Metal Resistant Endophytic Bacteria Reduces Cadmium, Nickel Toxicity, and Enhances Expression of Metal Stress Related Genes with Improved Growth of Oryza Sativa, via Regulating Its Antioxidant Machinery and Endogenous Hormones

**DOI:** 10.3390/plants8100363

**Published:** 2019-09-23

**Authors:** Rahmatullah Jan, Muhammad Aaqil Khan, Sajjad Asaf, In-Jung Lee, Kyung Min Kim

**Affiliations:** 1School of Applied Biosciences, Kyungpook National University, Daegu 41566, Korea; rehmatbot@yahoo.com (R.J.); aqil_bacha@yahoo.com (M.A.K.); ijlee@knu.ac.kr (I.-J.L.); 2Natural and Medical Science Research Center, University of Nizwa 616, Nizwa 611, Oman; sajadasif2000@gmail.com; 3Department of Botany, Garden Campus, Abdul Wali Khan University, Mardan 23200, Pakistan; Lubnabilal68@yahoo.com

**Keywords:** *Exiguobacterium indicum*, OsMTP1, phytohormone, synergistic, detoxification

## Abstract

The tolerance of plant growth-promoting endophytes (PGPEs) against various concentrations of cadmium (Cd) and nickel (Ni) was investigated. Two glutathione-producing bacterial strains (*Enterobacter ludwigii* SAK5 and *Exiguobacterium indicum* SA22) were screened for Cd and Ni accumulation and tolerance in contaminated media, which showed resistance up to 1.0 mM. Both strains were further evaluated by inoculating specific plants with the bacteria for five days prior to heavy metal treatment (0.5 and 1.0 mM). The enhancement of biomass and growth attributes such as the root length, shoot length, root fresh weight, shoot fresh weight, and chlorophyll content were compared between treated inoculated plants and treated non-inoculated plants. Both strains significantly increased the accumulation of Cd and Ni in inoculated plants. The accumulation of both heavy metals was higher in the roots than in the shoots, however; Ni accumulation was greater than Cd. Heavy metal stress-responsive genes such as *OsGST*, *OsMTP1*, and *OsPCS1* were significantly upregulated in treated non-inoculated plants compared with treated inoculated plants, suggesting that both strains reduced heavy metal stress. Similarly, abscisic acid (ABA) was increased with increased heavy metal concentration; however, it was reduced in inoculated plants compared with non-inoculated plants. Salicylic acid (SA) was found to exert synergistic effects with ABA. The application of suitable endophytic bacteria can protect against heavy metal hyperaccumulation by enhancing detoxification mechanisms.

## 1. Introduction

Heavy metals are high-density metallic elements that can cause toxicity at very low levels of exposure [1]. Some heavy metals are an essential part of nutrition (Fe, Co, and Zn) or harmless (Ru, Ag, and In); however, they can be harmful in excess amounts. On the other hand, some heavy metals are highly toxic even in small amounts, such as Cd, Hg, and Pb. The main sources of heavy metal contamination in the environment include industrial, agricultural, pharmaceutical, and domestic sources [2]. Wastewater irrigation of agricultural lands can result in considerable heavy metal accumulation and accelerate the uptake of heavy metals by plants, which would in turn reduce food quality and safety [3]. Due to the high accumulation of heavy metals in soil and plants, the risk on human health has increased in the past few decades; thus, heavy metals are potentially hazardous environmental pollutants [4]. Furthermore, anthropogenic activities continuously increase heavy metal pollution in both soil and aquatic environments. The resulting heavy metal toxicity in plants could negatively affect plant growth [5].

Heavy metals inhibit plant growth by suppressing different processes such as photosynthesis [6], respiration [7], carbohydrate metabolism [8], and water relation. It has also been reported that heavy metal accumulation in plants decreases growth parameters due to functional interference in the uptake and distribution of important mineral nutrients such as Ca, Mg, Fe, P, and K [5,8]. Among the heavy metals, Cd and Ni are the main environmentally hazardous metals and their accumulation in rice plants is well documented [9,10]. Usually, plants obtain Cd and Ni through the root system by active transport and passive diffusion; however, the accumulation of Cd and Ni in the leaf system has also been reported for sunflower, soybean, tomato, and oat leaves [11]. Exceeding the tolerable concentration of Cd and Ni causes toxicity in plants and can disrupt cellular functions due to metabolism inhibition, cellular injury, and sometimes plant death in severe cases. The toxicity in plants may be attributed to the replacement of a part of a molecule with a metal, obstruction of biologically important functional groups in a molecule, and alteration of plasma membrane enzymes [11]. Additionally, Cd causes cellular injury due to the generation of reactive oxygen species (ROS), resulting in the disruption of nucleic acid and protein synthesis [12,13]. Cd inhibits the enzymatic activity of nitrate reductase, which consequently inhibits the uptake of nitrate and its transport from the root to shoot [14], decreases nitrogen fixation [15], and inhibits the ATPase activity of the plasma membrane [16]. A high Ni concentration causes necrosis and chlorosis [17], inhibits CO_2_ fixation, and affects the Calvin cycle by altering enzymatic activity [18].

To cope with toxicity, plants possess a number of sophisticated mechanisms for heavy metal detoxification and scavenging, such as increasing tolerability by chelating heavy metals [19], regulating water potential disruption by heavy metals [20], osmotic regulation via homeostatic processes, and heavy metal uptake reduction [21]. Additionally, plants have antioxidant defense mechanisms that alleviate the toxicity of heavy metals using different enzymes such as Glutathione reductase (GR), Superoxide dismutase (SOD), Guaiacol peroxidase (GPX), and Catalase (CAT); furthermore, some nonenzymatic antioxidants, such as glutathione and ascorbate, act the same as enzymatic antioxidants [22]. Phytochelatins (PCs) are heavy metal-tolerating peptides produced mostly under Cd, Ni, Cu, Hg, and Pb stress [23,24]. In addition to these detoxification and scavenging mechanisms, plants also regulate specific genes responsible for resistance to toxic elements to counteract the stress stimuli. Various genes responsible for heavy metal stress have been reported in previous studies; however, we aim to investigate the most relevant genes responsible for Cd and Ni stress, such as *OsGST*, *OsMTP1*, and *OsPCS1*. Glutathione-*s*-transferase (GST) is a key detoxifying enzyme during Cd stress [25]; it catalyzes the nucleophilic conjugation of reduced tripeptide glutathione (γ-Glu-Cys-Gly) into various hydrophobic and electrophilic substrates [26], making them more soluble and easier to excrete [27,28]. PCs, which are synthesized by phytochelatin synthase 1 (PCS1), are a well-known example of metal-induced metabolite accumulation and are implicated in cellular metal detoxification [29]. The overexpression of PCS1 in transgenic tobacco and *Arabidopsis thaliana* has been reported to increase tolerance to and accumulation of Cd by elevating thiols and PCs [30,31]. PCs are not involved in the detoxification of all heavy metals; they are limited to a few metals such as Cd, Ni, Cu, and AsO_2_ [32]. Similarly, some plant transporter proteins, such as cation diffusion facilitator (CDF), are responsible for metal homeostasis, which mediate the movement of heavy metals across the membrane. These CDF proteins have been designated as metal tolerance proteins (MTPs). A previous study found that *OsMTP1* gene expression was significantly induced by Cd inoculation and its heterologous expression in yeast mutants restored Cd, Ni, and Zn tolerance [33]. Similar to gene expression, phytohormone regulation is a defense strategy of plants during biotic and abiotic stress. Phytohormones are small signaling compounds, which are involved to some extent in plant growth, development, and defense. Increasing evidence suggests that plant hormones, such as abscisic acid (ABA) and salicylic acid (SA), are involved in heavy metal stress signaling in plants [34]. In various plants, ABA and SA levels are increased with Cd stress; however, they can also increase Ni resistance [35,36,37,38].

Together with the plant defense system, plant growth-promoting endophytes (PGPEs) also play a vital role in the detoxification of heavy metals. Plants contribute to phytoremediation by direct involvement or enhancing detoxification activity through microbes [39]. Phytoremediation with microbes has attracted increasing attention for removing contaminants from the environment [40]. Among these microbes, endophytic bacteria can promote plant growth and decrease biotic and abiotic stress (especially heavy metal stress) without harming the host plant [41]. PGPEs have developed various mechanisms to eliminate heavy metal stress including the removal of metal ions from cells, conversion to less toxic forms, confiscation of metals on the cell surface, adsorption, biomethylation, and extracellular precipitation [42,43]. Recent studies have demonstrated that PGPEs could alter the function of plant antioxidant enzymes such as SOD, POS, CAT, ascorbate peroxidase, and glutathione peroxidase, which are part of the plant defense system produced in response to oxidative stress due to heavy metals [44,45]. Plant growth-promoting bacteria (PGPB) possess heavy metal resistance genes responsible for the detoxification and translocation of various heavy metals such as Cd, Ni, Hg, Zn, and Cu. In addition, some PGPEs enhance Cd and Ni accumulation in plants [46]. *E. ludwigii* SAK5 and *E. indicum* SA22 are glutathione-producing bacteria and important heavy metal accumulators and mobilizers, which encode several genes involved in the uptake and efflux of heavy metals including Ni [47,48] and Cd [49]. The present study was conducted to evaluate the role of PGPEs (*E. ludwigii* SAK5 and *E. indicum* SA22) in Cd and Ni accumulation, detoxification, and translocation and their effect on the growth attributes of rice plants.

## 2. Materials and Methods

### 2.1. Isolation, Screening, Bioassay, and Identification of Bacterial Endophytes 

Endophytic bacteria were isolated from various plants (*Artemisia princeps*, *Chenopodium ficifolium*, *Oenothera biennis,* and *Echinochloa crus-galli*) collected from Pohang Beach, South Korea, following a previous protocol [50,51]. Morphological features such as size, color, shape, and growth pattern on LB agar for 24 h of the isolated endophytes were examined. Furthermore, all isolates were screened for indole acetic acid (IAA) production using the method of Patten and Glick [52], siderophore production using the method of Schwyn and Neilands [53], and phosphate solubilization using the method of Katznelson and Bose [54]. The isolated endophytes were screened for growth-promoting effects in rice plants. Overall, six of sixteen endophytic bacteria significantly enhanced plant growth and biomass. The selected six endophytic strains with high growth-promoting capability were further screened for resistance to Cd and Ni. To determine the tolerance of the six strains to heavy metal stress, the growth of the strains were examined on (potato dextrose agar) PDA plates with different concentrations of Cd and Ni (0.5, 1.0, and 1.5 mM for each metal). The growth was assessed after incubation at 30 °C for one week in the dark [55]. Two of these isolates showed higher tolerance to Cd and Ni at concentrations of 0.5 and 1.0 mM; however, plant growth was affected at 1.5 mM. Based on this evaluation, the tolerable concentrations for both heavy metals were further investigated. Furthermore, as the two isolates could promote plant growth and heavy metal tolerance, they were selected for further analysis.

Identification was based on the sequencing of partial 16S ribosomal RNA (rRNA). According to the standard procedure of Sambrook and Russell [56], total DNA was isolated and the 16S rRNA gene was PCR-amplified using the 27F primer (5′-AGAGTTTGATC(AC)TGGCTCAG-3′) and 1492R primer (5′-CGG (CT) TACCTTGTTACGACTT-3′), complementary to the 5′ and 3′ ends of the prokaryotic 16S rRNA, respectively. The BLAST search program and EzTaxon-e were used to identify similar reference sequences homologous to the nucleotide sequences of bacterial isolates in this study. The most similar reference sequences with the highest homology and query coverage and the lowest E values were selected and aligned by ClustalW using MEGA version 6.0 software. The phylogenetic tree was constructed using the neighbor-joining method in MEGA version 6.0 software (Pennsylvania State University lab, Japan).

### 2.2. Quantification of In Vitro IAA by Gas Chromatography-Mass Spectrometry/Selective Ion Monitoring (GC-MS/SIM)

The bacterial strains were grown in LB media (10 g tryptone (DIFCO, USA), 5 g yeast (Becton, USA) extract, pH 7.0 ± 0.2) for three days, centrifuged at 5000× *g* for 10 min to separate the cells from the culture broth, and filtered through a 45 μm filter. The isolated culture filtrate was analyzed by GC-MS/SIM to determine the IAA content, following methods previously described [51,57]. The concentration of IAA in the broth was calculated by comparing the peak areas of IAA with those of the known standard by GC-MS/SIM.

### 2.3. Glutathione Production Assessment with Monochlorobimane (MCB)

To further confirm the tolerance of the selected bacterial strains to heavy metal stress, the strains were screened for glutathione production. Both strains were grown on PDA plates for 48 h and analyzed for glutathione production according to a previous protocol [58]. MCB (40 mM) was prepared by dissolving in acetonitrile (Sigma, Germany) protected from light and stored at −20 °C. To evaluate glutathione production, around 15 µL of MCB solution was sprayed on each bacterial plate and allowed to stand for 2 min before exposure to UV light (365 nm). Strains expressing glutathione showed intense fluorescence and were used for further experiments.

### 2.4. Hydroponic Experimental Setup and Growth Conditions

Rice seeds of Ilmi cultivar were obtained from the Plant Molecular Breeding Lab of Kyungpook National University (Daegu, South Korea) and were screened in hydroponics. The seeds were sterilized in 70% ethanol for 5 min and 2% sodium hypochlorite (NaOCl) (Sigma, Germany) for 30 min and washed three times with double distilled water. The seeds were kept in an incubator at 25 °C for three days for proper germination. Further experiments were conducted using hydroponic medium in magenta boxes according to a previous method [59]. The whole experiment was designed in three main groups: i.e control, treated with Cd, and treated with Ni. The 1st group consisted of control, SAK5, and SA22 inoculated plants; the 2nd group involved treatments with Cd stress; and the 3rd group involved treatment with Ni stress (see for detail Appendix A). After successful growth, the seedlings were transplanted into hydroponic medium (Hoagland solution, pH 5.8) in magenta boxes to assess their response to Cd and Ni accumulation. A total of 12 seedlings were placed in each box on a gauze pad supported with Styrofoam (ORUM, South Korea) containing 200 mL of Hoagland solution (MB Cell, South Korea) [60]. After 10 days of normal growth in Hoagland solution, specific plants were inoculated with bacteria for five days prior to heavy metal treatment to develop a symbiotic relationship between the plants and bacteria. Both strains were grown in LB liquid medium until the optical density (OD) reached 600, then spell down about 3–4 grams resuspended in 40 mL distilled water, and inoculated to the respective pot. All plants were kept in a controlled growth chamber at 25 °C with a 16 h light duration. The plants were grown for five weeks and all assessments were performed after harvesting.

### 2.5. Assessment of Growth Parameters under Cd and Ni Stress

To evaluate the effect of Cd and Ni on the growth of rice plants, different growth parameters were examined after 20 days of stress. The chlorophyll content was measured for three replicates by selecting the third leaf using a chlorophyll meter (SPAD-502 Minolta; Tokyo, Japan) [61]. At the end of the experiment, the root, shoot length, and fresh weight were calculated and the samples were further freeze-dried for Inductively coupled plasma (ICP), amino acid, and sugar content analyses. The RNA was extracted for the analysis of genes affected by heavy metal stress using young leaves, stems, and roots, which were immediately placed in liquid nitrogen after harvesting. The remaining samples were kept at −80 °C until further use.

### 2.6. RNA Isolation and qPCR Analysis of Selected Genes

Total RNA was isolated from leaves using the RNeasy® Plant Mini Kit (Qiagen, Valencia, CA, USA). The cDNA was synthesized from 2 μg of total RNA using the qPCR-Bio cDAN Synthesis Kit following the manufacturer’s instructions. The accession number and primers used for each gene are listed in Table 1. To identify the transcriptional level of each gene in response to heavy metal stress, qPCR was performed using the Illumina Eco System (Illumina, San Diego, CA, USA). The reaction conditions included initial denaturation at 94 °C for 2 min, 40 PCR cycles at 94 °C for 10 s and 55 °C for 30 s, and 60–95 °C for amplicon specificity verification. Actin was used as housekeeping gene.

### 2.7. Cd and Ni Uptake Analysis by Inductively Coupled Plasma Mass Spectrometry (ICP-MS)

At the end of the experiment, the plants roots and shoots in each treatment group were harvested and washed in deionized water to prevent the superficial adhesion of metals. To further analyze the Cd and Ni content in the roots and shoots, the samples were freeze-dried and crushed into fine powder in liquid nitrogen. Subsequently, dried samples (200 mg each) were dissolved in HNO_3_ and HClO_4_ (4:1 *v/v*) (Sigma, Germany) solution digested in a microwave at 180 °C for 20 min. The prepared samples were quantified by ICP-MS (Optima 7900DV; PerkinElmer, Waltham, MA, USA) using external calibration method. External standard of known concentration was used to generate a calibration curve of instrument response. The generated calibration curve was used to back-calculate the concentration of unknown analyte in the samples, based on their instrument response. Instrument settings were adjusted to optimize the collective signal strengths of both the metals. The test solutions employed several different perturbed conditions, i.e., sample delivery rate of 0.6 mL min^−1^ and 1.9 mL min^−1^; torch positions of 6 and 12 mm; 5% CH_3_COOH; and 10 ppm NaCl matrix [62].

### 2.8. ABA and SA Accumulation under Cd and Ni Stress

To quantify plant endogenous ABA and SA content, the ABA and SA of inoculated and non-inoculated samples were extracted from freeze-dried samples (three replicates). The quantification and extraction of ABA were performed using a previous protocol [63,64] with some modifications. Around 150 mg of powder was homogenized in 2 mL of 90% methanol including 15 mg of butylated hydroxytoluene and 20 mL of 2% glacial acetic acid. The homogenate was incubated for 48 h at 4 °C, dried using a rotary evaporator, and methylated with diazomethane for further analysis. ABA was quantitatively assessed by GC-MS/SIM (5973 Network Mass Selective Detector and 6890N Network GC System; Agilent Technologies, Palo Alto, CA, USA) in three identical repeats.

The lyophilized sample was further crushed into fine powder in liquid nitrogen for SA quantification following a previous method [65]. Additionally, the powdered sample (0.2 g) was mixed with 2 mL of 90% methanol (Sigma, Germany) and centrifuged for 20 min at 10,000× *g*. The methanol in the supernatant was evaporated in a vacuum centrifuge and the sample was resuspended in 3 mL of 5% trichloroacetic acid (Sigma, Germany). The upper organic layer was further mixed with a solution of isopropanol, ethyl acetate, and cyclopentane (1:49.5:49.5 *v/v*) (Duksan, South Korea) and vigorously vortexed. The uppermost layer was transferred to a 4 mL tube and vacuum dried. Prior to high-performance liquid chromatography (HPLC), the dried pellet was mixed with 1 mL of HPLC mobile phase and SA was quantified through fluorescence detection.

### 2.9. Free Amino Acid (FAA) Quantification

To quantify FAAs, the fresh sample (0.5 g) was crushed in liquid nitrogen and extracted by shaking in 10 mL of 70% methanol for 24 h. The FAA content was determined using the EZ: faast analysis kit (Phenomenex, Santa Clara, CA, USA) according to the manufacturer’s instructions. The FAA content was analyzed by GC-MS using a Hewlett-Packard (HP) 6890N/5975 instrument (Agilent Technologies, Torrance, CA, USA) and a ZB-AAA (10 m × 0.25 mm) amino acid analysis column with a constant carrier gas flow and an oven temperature program as described by Pavlik et al. [66].

### 2.10. Estimation of Soluble Sugars

The soluble sugar content was estimated using lyophilized foliar tissues (500 mg) powdered in liquid nitrogen and homogenized with 80% ethanol (2 mL) at 80 °C for 20 min [67]. The homogenate was centrifuged for 15 min at 10,000 rpm and the pellet was resuspended in 4 mL of double distilled water and filtered with a 0.2 mm pore filter. Glucose, fructose, and sucrose were measured by HPLC equipped with Bio-Rad Aminex 87C column (300 × 7.8 mm); water was used as an eluent with a flow rate of 0.6 mL/min.

### 2.11. Statistical Analysis 

Experiments were performed in triplicate and the values obtained are presented as the mean ± standard error. The data were statistically evaluated by Duncan’s multiple range test (DMRT) using GraphPad Prism (version 6.01; San Diego, CA, USA).

## 3. Results

### 3.1. Screening of Endophytic Bacteria for Heavy Metal Stress and Glutathione Biosynthesis

Among the six endophytic bacterial strains, only SAK5 and SA22 were highly tolerant to Cd and Ni and showed a normal growth pattern in Cd-supplemented and Ni-supplemented PDA media, whereas the other four endophytes were sensitive to Cd and Ni (Appendix A) (their growth was inhibited by these metals). Both strains were more tolerant to 0.5 mM than 1.0 mM Cd and Ni, and above this concentration (i.e., 1.5 mM), growth was inhibited (Figure 1A). Control (0 mM Cd and Ni) of both the SA22 and SAK5 are shown in Appendix A. Furthermore, in comparison with the SA22 strain, the SAK5 strain showed greater tolerance to both metals. Based on the results, both strains were further analyzed by growing them in potato dextrose broth (PDB) supplemented with 0.5 mM Cd and Ni separately to assess their efficiency in accumulating heavy metals by ICP-MS quantification. Our results confirmed that both strains were highly efficient in accumulating both heavy metals (Figure 1B). However, compared with SA22, SAK5 showed the best result and accumulated more Cd and Ni. Cd accumulation in SAK5 was 31% higher than in SA22 and Ni accumulation in SAK5 was 33% higher than in SA22. Furthermore, Cd accumulation was 3% more than Ni in SAK5. In addition to heavy metal uptake analysis, both strains were analyzed for glutathione production considering their active participation in stress inhibition. All endophytes sprayed with MCB showed fluorescence when exposed to UV light (365 nm). Although fluorescence was observed for both strains (positive strains), the intensity was slightly different, which indicated variation in the production of glutathione. We believe that the high UV light intensity may be attributed to the high production of glutathione as it conjugates with the MCB substrate and activates GST, which is an enzyme responsible for glutathione biosynthesis [58]. For further experiments, we selected both strains (SAK5 and SA22), which are considered the strains with the highest glutathione-producing ability (Figure 1C).

Furthermore, the SAK5 and SA22 strains were molecularly identified through 16S rRNA gene sequencing, which revealed that both endophytic bacteria showed sequence identity with *E. ludwigii* and *E. indicum,* respectively. Similarly, BLAST analysis indicated that the SAK5 and SA22 bacterial strains (GenBank accession number: MK834790 and MG706139) were 99% similar to *Enterobacter ludwigii* SAK5 and *Exiguobacterium indicum* SA22, respectively. Phylogenetic analysis indicated that SA22 formed a clade with *E. indicum*, whereas SAK5 formed a clade with *E. ludwigii* with high bootstrap values (Appendix A).

### 3.2. IAA Quantification in the Culture Broth of Selected Isolates

The cultural filtrates of selected endophytic bacterial isolates were analyzed for the quantification of IAA by GC-MS/SIM. Our results showed that both of the selected endophytic strains (SAK5 and SA22) were able to produce significant amounts of IAA (Figure 1D). The highest amount of IAA was produced by the bacterial isolate *E. ludwigii* SAK5 (2.7 ± 0.7 µg/mL), followed by *E. indicum* SA22 (0.14 µg/mL).

### 3.3. Evaluation of Phenotypic Attributes and Chlorophyll Content under Cd and Ni Stress

In the present study, rice seedlings were inoculated with the SAK5 and SA22 endophytes separately, with or without Cd and Ni treatment (0.5 and 1.0 mM), under hydroponic conditions. After one month, a difference was observed in the growth parameters of the inoculated and non-inoculated seedlings (Figure 2). The effects of the inoculation on rice seedlings under Cd and Ni stress were evaluated by measuring the root shoot length, root shoot fresh/dry weight, and chlorophyll content (Figure 3). The results indicated that both endophytes promoted growth and increased the plant biomass and chlorophyll content. In the SAK5-inoculated and SA22-inoculated nontreated plants, the shoot length was increased by 27.2% and 13.6%, respectively, and the root length was increased by 24.1% and 13.7%, respectively. Cd and Ni treatment significantly decreased growth attributes with increasing concentration. All phenotypic traits of growth were higher with 0.5 mM than 1.0 mM Cd and Ni for inoculated and non-inoculated plants. Based on screening results, 1.0 mM of both heavy metals decreased the growth of both SAK5 and SA22; thus, all parameters were decreased at 1.0 mM. A comparison of the shoot and root lengths of 0.5 mM Cd-treated plants without endophyte inoculation and those of Cd-treated plants inoculated with either strain showed that the SAK5 and SA22 strains increased the shoot length by 30% and 40%, respectively, and root length by 23.8% and 18.5%, respectively. In the case of Ni-treated inoculated and non-inoculated plants, the shoot and root lengths of SA22-inoculated plants were greatly increased compared with those of SAK5-inoculated plants. The same pattern was observed for the shoot fresh weight (SFW), shoot dry weight (SDW), root fresh weight (RFW), and root dry weight (RDW) in almost all the treatment groups (Figure 3). Cd and Ni contamination adversely decreased the chlorophyll content; however, both endophytes markedly increased the chlorophyll content. In group 1, the chlorophyll content was increased by 24% in SAK5-inoculated plants and 26% in SA22-inoculated plants compared with control plants. In group 2, the chlorophyll content was lower in 1.0 mM Cd-treated plants than in 0.5 mM Cd-treated plants, with or without endophyte inoculation; nevertheless, it was higher in plants inoculated with either strains compared with plants without inoculation. A similar pattern was observed in group 3; however, the enhancement was higher in Ni-treated plants than in Cd-treated plants.

### 3.4. Gene Regulation under Heavy Metal Stress

Our results revealed that *OsGST* expression was similar between inoculated nontreated plants and control plants due to the lack of stress (Figure 4A). The expression level was increased with increased stress. The results showed that *OsGST* was highly upregulated in treated non-inoculated plants with 1.0 mM Cd and Ni; however, the expression was decreased when treated with 0.5 mM Cd and Ni. In the treated inoculated plants, the expression level was also increased with an increase in the concentration of heavy metals. The expression level was lower in treated inoculated plants than in treated non-inoculated plants; however, *OsGST* was highly upregulated in Ni-treated plants compared with Cd-treated plants. In the present study, the *OsMTP1* gene was also evaluated and the result was similar to the expression pattern of *OsGST*. *OsMTP1* was highly upregulated in non-inoculated plants treated with 1.0 mM Cd and Ni followed by 0.5 mM Cd and Ni (Figure 4B). The results demonstrated that the expression level was higher in treated non-inoculated plants than in treated inoculated plants. Furthermore, the expression level was higher in inoculated plants treated with 1.0 mM Cd and Ni than in inoculated plants treated with 0.5 mM Cd and Ni. Therefore, *OsMTP1* expression may be directly proportional to heavy metal stress.

Moreover, together with the *OsGST* and *OsMTP1* genes, other genes such as *OsPCS1* could also be upregulated during the hyperaccumulation of heavy metals and enhance the detoxification of heavy metals by increasing PC levels. The results revealed that the expression of *OsPCS1* was similar to that of the other two genes. The expression level was increased with increasing heavy metal concentration. However, *OsPCS1* upregulation was greater in treated non-inoculated plants than in treated inoculated plants (Figure 4C). These results indicated that both strains decreased the stress caused by heavy metals due to their active participation in detoxification.

### 3.5. Cd and Ni Accumulation and Translocation to Shoots

Rice plants have considerable potential for heavy metal accumulation and translocation, especially for Cd and Ni. The accumulation of both heavy metals in the roots and shoots of rice seedlings is shown in Figure 5. ICP-MS analysis demonstrated that Cd and Ni accumulation and translocation were not uniform and a huge fraction remained in the roots, as reported previously [68]. However, Cd and Ni were not detected in the roots and shoots of group 1 (control, SAK5-inoculated, and SA22-inoculated nontreated plants). The capacity for the accumulation of both heavy metals was increased at 0.5 mM in both the inoculated and non-inoculated plants, as compared to 1.0 mM; however, it was greater in inoculated plants than in non-inoculated plants, which indicated that both endophytes were responsible for heavy metal accumulation. Cd accumulation in both the shoots and roots was higher in treated plants inoculated with either strain compared with treated plants without inoculation (Figure 5A,B). Interestingly, the accumulation of Cd in the roots was lower at 1.0 mM than at 0.5 mM. Our results revealed that Cd accumulation in the shoots at 0.5 mM was 72.4% and 145.8% higher in plants inoculated with SAK5 and SA22, respectively, than in non-inoculated plants. In addition, Cd accumulation in the roots was increased by 310% and 310.3% in plants inoculated with SAK5 and SA22, respectively.

The evaluation of both heavy metals showed that Ni accumulation was higher than Cd accumulation in both the shoots and roots. Similar to Cd accumulation, Ni also demonstrated the same pattern of accumulation and the concentration was decreased with increased concentration in the hydroponic solution. Interestingly, the concentration of Ni accumulated in the roots of non-inoculated plants treated with 1.0 mM Ni (group 3) was higher than the concentration of Cd accumulated in the roots of plants inoculated with either endophyte and treated with 0.5 mM Cd (group 2). Furthermore, our results demonstrated that Ni accumulation in the roots of non-inoculated plants was 68.9% higher at 1.0 mM than at 0.5 mM. Comparative analysis revealed that Ni accumulation in the shoots at 0.5 mM was significantly increased by 163% and 155% in SAK5-inoculated and SA22-inoculated plants, respectively, compared with non-inoculated treated plants. In addition, Ni accumulation in the roots was increased by 340% and 300% in SAK5-inoculated and SA22-inoculated plants, respectively (Figure 5C,D).

### 3.6. Regulation of ABA and SA under Cd and Ni Stress Coupled with PGPB

The result of ABA and SA quantification in rice plants under heavy metal stress is shown in Figure 6. In the present study, the symbiotic behavior of the SAK5 and SA22 strains with rice plants significantly increased the ABA level in Cd-treated and Ni-treated plants compared with control plants (Figure 6). In comparison with control plants, plants inoculated with either strain without treatment with heavy metals (group 1) showed reduced ABA levels. However, in comparison with 0.5 mM Cd-treated non-inoculated plants, Cd-treated plants inoculated with endophytic bacteria (group 2) exhibited significantly lower ABA levels (16% and 13% in SAK5-inoculated and SA22-inoculated plants, respectively). Similarly, when the concentration was increased to 1.0 mM, the level of ABA was enhanced compared with the level at 0.5 mM; however, it was significantly reduced in SAK5-inoculated and SA22-inoculated plants (16% and 20%, respectively) compared with treated non-inoculated plants at 1.0 mM. In addition, the ABA level was greatly decreased in Ni-treated plants (group 3) compared with Cd-treated plants. Although the pattern was similar to that under Cd stress, the decrease was lower in SAK5-inoculated plants (4% and 6% at 0.5 and 1.0 mM, respectively) than in SA22-inoculated plants (56% and 38% at 0.5 and 1.0 mM, respectively). These findings demonstrated the interaction of both PGPB with rice plants, which facilitated the alleviation of Cd and Ni toxicity in the plants.

Unlike ABA, SA was decreased significantly under heavy metal stress, which confirmed the antagonistic effect between SA and ABA. Our results showed that SA was highly upregulated in SA22-inoculated nontreated, SAK5-inoculated nontreated, and control plants (Figure 6B). However, the SA level was lower in almost all Cd-treated and Ni-treated plants compared with control and endophyte-inoculated non-treated plants. Under Cd stress, treated non-inoculated plants exhibited lower SA levels compared with treated inoculated plants; however, an increase in heavy metal concentration led to the downregulation of SA. Although the same pattern was observed under Ni stress, the decreasing ratio was higher than that under Cd stress. In comparison with 0.5 and 1.0 mM Ni-treated plants without inoculation, Ni-treated plants inoculated with SAK5 showed a 18% and 24% increase in SA, respectively, and Ni-treated plants inoculated with SA22 showed a 23% and 29% increase in SA, respectively. SA is downregulated during heavy metal stress; however, our study revealed the upregulation of SA during interaction with PGPB, which indicated that these endophytes could enhance the plant defense system to reduce heavy metal stress.

### 3.7. Heavy Metal Stress Regulation of Proline Biosynthesis

In the present study, proline was quantified by GC-MS, which confirmed that increasing the concentration of both Cd and Ni also significantly increased the accumulation of proline at an optimum concentration (0.5 mM); however, it adversely affected proline accumulation at an intolerable concentration (1.0 mM) due to the inhibition of certain physiological processes (Figure 7A). Between the two heavy metals, Ni is a stronger regulator of proline biosynthesis at 0.5 mM. Our results showed that the proline concentration was lower in control, SAK5-inoculated, and SA22-inoculated plants without heavy metal treatment (group 1) than in SAK5-inoculated and SA22-inoculated plants treated with heavy metals. In group 1, a significant increase was observed in SAK5-inoculated and SA22-inoculated plants (36.2% and 43.5%, respectively) compared with control plants. In group 2, in comparison with 0.5 mM Cd-treated plants without inoculation, Cd-treated plants inoculated with SAK5 and SA22 had significantly increased proline levels (49.9% and 28.6%, respectively). Although a decreasing pattern was observed with Cd treatment at 1.0 mM compared with 0.5 mM, proline was increased in Cd-treated plants inoculated with either strains compared with Cd-treated non-inoculated plants. A similar pattern was observed for plants treated with Ni (group 3); however, proline levels were higher in Ni-treated plants than in Cd-treated plants. The present study found that exposure to a higher concentration of Ni (1.0 mM) significantly decreased proline accumulation in plants. Both strains decreased heavy metal stress in 0.5 mM Ni-treated plants, which increased proline accumulation by 47% and 29% in SAK5-inoculated and SA22-inoculated plants, respectively, compared with non-inoculated plants. Comparative analysis of both endophytes revealed that SAK5 was associated with a higher efficiency of proline accumulation. These results suggest that the SAK5 and SA22 endophytes are stress inhibitors and may reduce heavy metal stress by regulating proline biosynthesis.

### 3.8. Quantitative Analysis of Sugar Content

Heavy metal stress can alter various physiological processes, which disrupts plant growth. Sugars are important constituents that facilitate several physiological processes during abiotic stresses. Fructose, sucrose, and glucose accumulation plays a vital role in osmoprotection, homeostasis, and free radical scavenging under stress conditions [69]. In the present study, fructose, sucrose, and glucose accumulation was evaluated under Cd and Ni stress. Our results demonstrated that glucose, fructose, and sucrose accumulation was enhanced in SAK5-inoculated (39%, 88%, and 83%, respectively) and SA22-inoculated (81%, 103%, and 92%, respectively) plants compared with control plants. Furthermore, we found that increasing the Cd and Ni concentration from 0.5 mM to 1.0 mM significantly enhanced the accumulation of all three sugars in treated non-inoculated plants. However, the accumulation of glucose and sucrose was similar in treated non-inoculated plants, whereas that of fructose was higher. In comparison with Cd-treated and Ni-treated non-inoculated plants, Cd-treated and Ni-treated inoculated plants had a reduced sugar content, which demonstrated heavy metal stress inhibition by the endophytes. Furthermore, in treated inoculated plants, sugar accumulation was significantly increased at a higher concentration of heavy metals, which indicated that a higher concentration of Cd and Ni could inhibit endophyte growth. Quantitative analysis revealed that highest concentration of fructose (followed by glucose and sucrose) was found in all non-inoculated plants treated with 1.0 mM Cd and Ni (Figure 7B–D).

## 4. Discussion

Heavy metal stress is a severe problem caused by natural and anthropogenic activities. Increasing the accumulation of heavy metals such as As, Cd, Pb, Ni, Cr, and Zn in plant cells could trigger physiological and biochemical processes, which result in severe damage to the plants and lead to plant death [39,70,71,72]. In various studies, endophytic bacteria such as *Paenibacillus* sp., *Bacillus* sp., *Exiguobacterium* sp., *Alcaligenes* sp., *Pantoea* sp., *Brevibacillus* sp., and *Pseudomonas* sp. have been reported to be highly efficient in accumulating Cd, Ni, and other heavy metals [73,74,75]. In the present study, we evaluated the heavy metal bioremediation potential of six different endophytic bacterial strains in Cd-supplemented and Ni-supplemented media. The initial analysis revealed that the *E. ludwigii* SAK5 and *E. indicum* SA22 strains were highly resistant to Cd and Ni contamination due to their enhanced growth rate when exposed to 0.5 and 1.0 mM Cd and Ni. Growth and ICP analyses confirmed that both strains were effective in Cd and Ni uptake; however, the uptake rates of SAK5 were higher (631 and 610 µg/mL for Cd and Ni, respectively) than those of SA22 (430 and 407 µg/mL for Cd and Ni, respectively), which demonstrated the high resistance and rehabilitation potential of *E. ludwigii* for both metals. The intercellular accumulation of both heavy metals by both strains suggests their capability in bioremediation, which is representative of the adaptation of endophytic bacteria to heavy metal toxicity. Our results revealed that the selected plant growth-promoting endophytic bacteria significantly enhanced the growth attributes of treated plants. In comparison with control plants, plants inoculated with either strain had the highest root shoot length, root shoot fresh/dry weight, and chlorophyll content; however, growth attributes were lower in Cd-treated and Ni-treated inoculated plants compared with treated non-inoculated plants [76]. Similar results were recently reported in another study [73], showing that SA22 regulated plant growth attributes under normal conditions and inhibited all growth parameters when exposed to Cd stress; however, the root shoot length and root shoot fresh/dry weight were comparatively higher than those of treated non-inoculated plants. In addition, our findings revealed that Ni reduced growth parameters and dry biomass, similar to the results for *Glycine max* [77]. In the present study, we investigated the glutathione production of selected bacterial strains using the MCB detection method (Figure 1C), which revealed a high production level in both strains. Glutathione can reduce heavy metal stress by eliminating peroxidase, regulating the cell cycle, and removing reactive species [78]. A previous study showed that the rapid production of ROS or oxidative burst caused by abiotic stress [79] significantly reduced the plant biomass due to the disruption of physiological and biochemical processes. Previous studies also reported that glutathione may be involved in the ROS defense mechanism [80], detoxification of heavy metals [81], and sequestration of xenobiotics [82]. Our results are consistent with those of previous studies showing the hypersensitivity of glutathione-deficient *Arabidopsis* mutants to both Cd and Cu [83,84]. Moreover, the upregulation of glutathione synthesis in various plants could enhance Cd and Ni tolerance and accumulation in the shoots [81,85].

Plants activate signaling pathways to counteract incoming stress signals by activating specialized genes, which trigger their corresponding enzymes to reduce stress. In this study, we evaluated *GST*, *MTP*, and *PCS* by qPCR, which are genes reported to be closely associated with Cd-mediated and Ni-mediated stress. *GST* is a secondary antioxidative enzyme dependent on the reduced glutathione molecule responsible for defense against the destructive effects of ROS. In a previous study [86], rice roots subjected to Cd stress (100 μM, 24 h) showed increased *GST* and *APX* gene expression. Three GST genes (*OsGSTU3*, *OsGSTU4*, and *OsGSTU12*) that belong to the plant-specific tau class were found to be markedly overexpressed in roots following Cd exposure. Another study [87] reported the strong induction of the *OsGST* gene (*OsGSTU5*) in response to Cd stress. GSTs are involved in the direct quenching of Cd ions, forming glutathione-Cd complexes [88]. Our results demonstrated that *OsGST* was highly upregulated under Cd and Ni stress. Previous studies have found that increasing the concentration of heavy metals (Cd and Ni) generates ROS [89,90], leading to oxidative burst and increasing *OsGST* expression. Therefore, it is possible that during Cd and Ni stress, ORS was produced, which induced *OsGST* in treated non-inoculated plants; on the other hand, the expression level was decreased during interaction with either endophyte (Figure 4A). These findings suggest that *E. ludwigii* SAK5 and *E. indicum* SA22 efficiently reduced the stress caused by heavy metals; as a result, the expression level of *OsGST* was reduced in treated inoculated plants compared with treated non-inoculated plants.

Under heavy metal stress, plants transcriptionally regulate not only single genes such as *OsGST,* but also *OsMTP1*. It is a member of the CDF protein family, which plays a key role in cation homeostasis in organisms.

The *OsMTP1* gene has been reported to be upregulated due to stress from Cd exposure [33]. qPCR analysis revealed that the expression of *OsMTP1* was lower in control and nontreated inoculated plants than in treated inoculated plants and it was highly upregulated in treated non-inoculated plants (Figure 4B). These results suggest that both Cd and Ni could increase the transcriptional level of *OsMTP1*. However, the selected strains were also found to reduce heavy metal stress, resulting in the inhibition of *OsMTP1*. A study demonstrated that the upregulation of *OsMTP1* was triggered by Cd stress; overexpression and silencing analysis confirmed its role in the transportation of Zn, Ni, and Cd, which is consistent with the result obtained from the functional evaluation of yeast mutants [33]. Quantitative trait locus (QTLs) localization has also identified the *OsMTP1* gene as a highly significant gene for enhancing Fe and Zn concentration in seeds [91]. These findings suggest that MTPs are responsible for the inhibition of heavy metal stress by isolating metal ions.

Chelation is a common reaction for the detoxification of heavy metals using specific metal-binding ligands such as glutathione. In a previous study, a glutathione-deficient mutant of *Arabidopsis* showed PC deficiency, which was highly sensitive to Cd stress [84], indicating that *PCS1* may be an active participant in heavy metal stress. Our results revealed that the *OsPCS1* gene was significantly upregulated when the plants were exposed to Cd and Ni (Figure 4C). The level of expression was increased with an increase in heavy metal concentration. In comparison with heavy metal-treated non-inoculated plants, heavy metal-treated inoculated plants exhibited lower expression levels. These results demonstrated that the expression of *OsPCS1* significantly induced heavy metal tolerance. The detoxification properties of *PCS1* have been confirmed in tomato plants via exposure to heavy metals [92]. The cloning of *OsPCS1*, *TaPCS1*, *AtPCS1*, *BjPCS1*, and *CePCS1* has revealed that the overexpression of these genes could efficiently increase heavy metal tolerance and accumulation [93]. In the present study, in inoculated plants, bacteria enhanced glutathione levels, which could provide a substrate for *OsPCS1* and enhance the accumulation of, and tolerance to, Cd and Ni.

In this study, we found higher heavy metal accumulation in the roots than in the shoots, which would reduce the risk of heavy metal accumulation in seeds and increase food security. The roots were in direct contact with the heavy metals, which enhanced heavy metal uptake as reported previously [94]. Interestingly, heavy metal uptake was significantly higher in treated inoculated plants than in treated non-inoculated plants, which demonstrated the active involvement of both strains in heavy metal uptake. This hyperaccumulation could be attributed to the mobilization of heavy metals due to biosurfactants produced by plant growth-promoting endophytic bacteria [95]. Previous studies have reported that *Enterobacter sp*. could significantly increase the accumulation of Cd, Cu, Co, Zn, and Pb [96] and enhance Zn, Cd, and Cu uptake when inoculated with *Sinapis alba* L. plants [97]. In addition, *Exiguobacterium* sp. could enhance the accumulation of Cd and Ni [98]. Based on our results, increasing the concentration of heavy metals adversely affected bacterial growth, resulting in the reduction of heavy metal uptake and other growth parameters.

To evaluate the role of *E. ludwigii* SAK5 and *E. indicum* SA22 in facilitating crosstalk during heavy metal stress, we investigated the accumulation of ABA and SA in plants. Inoculation with both strains significantly suppressed ABA accumulation, suggesting the role of both strains as stress inhibitors (Figure 6A). ABA accumulation has been reported to increase during Cd stress in *Phragmites australis* and rice plants [35] and a similar result was observed for crowberries treated with Cu and Ni [99]. On the other hand, ABA was increased in treated non-inoculated plants with increasing heavy metal concentration. In contrast to endogenous ABA, SA was decreased in treated non-inoculated plants; however, enhancement was observed in inoculated plants compared with non-inoculated plants. Similar to a previous study [100], SA was increased in heavy metal-treated plants inoculated with endophytic bacteria. These results suggest that both hormones may have synergistic effects on heavy metal stress, as observed in another study [101]. SA elevation and ABA reduction may be attributed to heavy metal stress inhibition by *E. ludwigii* SAK5 and *E. indicum* SA22. ABA may be upregulated by water loss during heavy metal stress [102]; ABA has been reported to reduce water loss through stomatal closure by activating stress-related genes [103,104]. Similar results were reported for white bean plants under Cd stress, which showed ABA enhancement and stomatal resistance [105].

In the current study, proline was quantified during heavy metal stress considering that previous studies reported its involvement in osmoregulation, enzyme denaturation prevention, cytosolic acidity regulation, hydroxyl radical scavenging, and injury [106]. The accumulation of proline was significantly higher in treated inoculated plants at 0.5 mM Cd and Ni and was reduced at 1.0 mM Cd and Ni (Figure 7A). Its accumulation was significantly reduced in treated non-inoculated plants, which could be attributed to the adverse effect of high Cd and Ni concentrations on physiological processes. In agreement with our results, a previous study found that the PGPB *Arthrobacter* sp. and *Bacillus* sp. significantly stimulated proline accumulation in pepper plants [107]. A recent study demonstrated that proline was increased in Ni-treated plants inoculated with *Enterobacter* sp. compared with Ni-treated non-inoculated plants and higher concentrations were found in the roots due to the higher accumulation of Ni in the roots [108]. The enhanced accumulation of proline detected in inoculated plants suggests the role of *E. ludwigii* SAK5 in stress mitigation in plants growing under Ni stress [109]. Furthermore, the reduced accumulation of proline in treated inoculated plants at a higher concentration (1.0 mM) may be attributed to the inhibited growth of the inoculated strains at a higher concentration.

Sugar accumulation has been found to increase during heavy metal stress and a higher accumulation has been observed under Cd and Zn stress than under Ni and Pb stress [110]. In the current study, we evaluated glucose, fructose, and sucrose accumulation in plants inoculated with endophytes under Cd and Ni stress. The content of all three sugars was significantly enhanced in endophyte-inoculated nontreated plants compared with control plants (Figure 7B–E), similar to the results of Islam et al., [111]. Sugar content was higher in treated non-inoculated plants than in treated inoculated plants. A previous study reported that Cd and Pb stress increased the sugar content in *Lemna polyrrhiza*; however, the highest heavy metal concentration and extended exposure decreased the sugar content [112]. Similar results were reported by Ahmad et al. [113], who suggested that an optimum salt concentration increased the sugar content in *P. sativum* plants, which decreased with increasing concentration; this may be attributed to a reduced photosynthesis rate or enhanced respiration rate. Our results revealed an increase in the sugar content when the heavy metal concentration was increased from 0.5 mM to 1.0 mM. In addition, our results demonstrated that the accumulation rate of all three sugars was significantly reduced in inoculated plants, compared with non-inoculated plants treated with heavy metals. However, the concentration of sugars in treated inoculated plants was higher at 1.0 mM than at 0.5 mM, which indicated that both endophytes significantly reduced heavy metal stress. Furthermore, sucrose has been reported as a main product of photosynthesis and a crucial sugar in plant life, involved in growth and development as well as signaling and stress inhibition.

## 5. Conclusions

The mutualistic relationship of PGPE with host plants could contribute to plant growth enhancement, biological control, and bioremediation. In this study, Cd and Ni exposure exerted stress, which inhibited growth and reduced biomass. Our results suggest that *E. ludwigii* SAK5 and *E. indicum* SA22 inoculation could efficiently enhance growth parameters under Cd and Ni stress. PGPE not only alleviate heavy metal stress by detoxification but also enhance the physiological and biochemical processes of plants, resulting in increased stress-inhibiting hormones by upregulating certain genes. These bacteria also enhance glutathione, proline, and sugar content, which are essential abiotic stress-reducing agents. Although these bacteria may be essential for phytoremediation and plant growth promotion, further studies are needed to understand the mechanism underlying the effects of PGPE on host plant growth under heavy metal stress to develop biotechnological applications based on essential plant–bacterial interactions.

## Figures and Tables

**Figure 1 plants-08-00363-f001:**
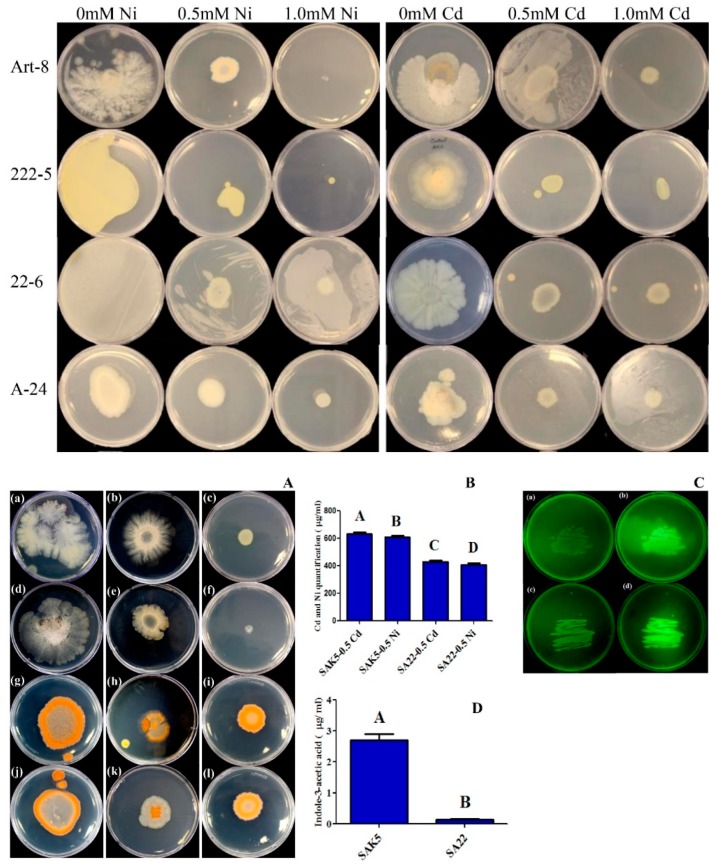
(**A**) Growth pattern of selected endophytic isolates in 0.5, 1.0, and 1.5 mM Cd-supplemented and Ni-supplemented media. The strains *Enterobacter ludwigii* SAK5 and *Exiguobacterium indicum* SA22 were screened for Cd and Ni toxicity and their growth rates were evaluated after one week of incubation at 30 °C. The A, B, and C plates represent the SAK5 strain grown with 0.5, 1.0, and 1.5 mM Cd, respectively, and the D, E, and F plates show the same strain grown with 0.5, 1.0, and 1.5 mM Ni, respectively. Similarly, the G, H, and I plates represent the SA22 strain grown with 0.5, 1.0, and 1.5 mM Cd, respectively, and the J, K, and L plates show the growth of the same strain with 0.5, 1.0, and 1.5 mM Ni, respectively. (**B**) ICP-MS analysis of Cd and Ni uptake by the SAK5 and SA22 strains. (**C**) Detection of glutathione production. A and B represent the SAK5 strain and C and D represent the SA22 strain. Both strains were grown on PDA media for 48 h at 30 °C and 40 mM monochlorobimane (MCB) was sprayed on the B and D plates, which were observed under UV light (365 nm) after 2 min. (**D**) Screening of the SKA5 and SA22 strains for IAA production. Concentration of IAA was calculated in µg/ml of culture filtrate. Data are means of three replicates along with standard error bars. Mean bars labeled with different letters are significantly different (*p* < 0.05) as evaluated by DMRT analysis.

**Figure 2 plants-08-00363-f002:**
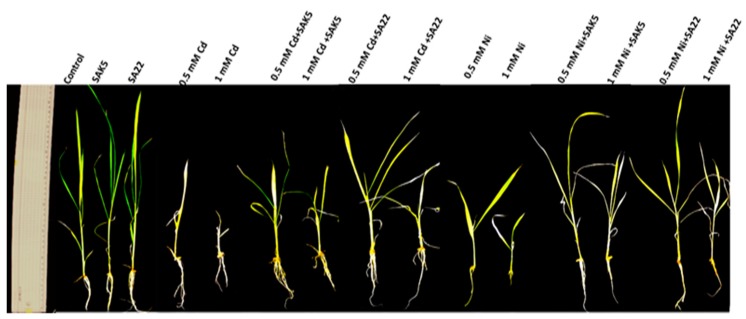
Effect of SAK5 and SA22 on the growth of rice seedling, under control conditions, cadmium (0.5 mM and 1 mM Cd), and nickel (0.5 mM and 1 mM Ni) stress. Control, untreated plants; SAK5 *Enterobacter ludwigii*-treated; SA22 *Exiguobacterium indicum*-treated; cadmium (0.5 mM and 1 mM)-treated plants; 0.5 mM and 1 mM Cd + SAK5, cadmium + SAK5-treated plants; 0.5 mM and 1 mM Cd + SA22, cadmium + SA22-treated plants; nickel, (0.5 mM and 1 mM)-treated plants; 0.5 mM and 1 mM Ni + SAK5, nickel + SAK5-treated plants; 0.5 mM and 1 mM Ni + SA22, nickel + SA22-treated plants.

**Figure 3 plants-08-00363-f003:**
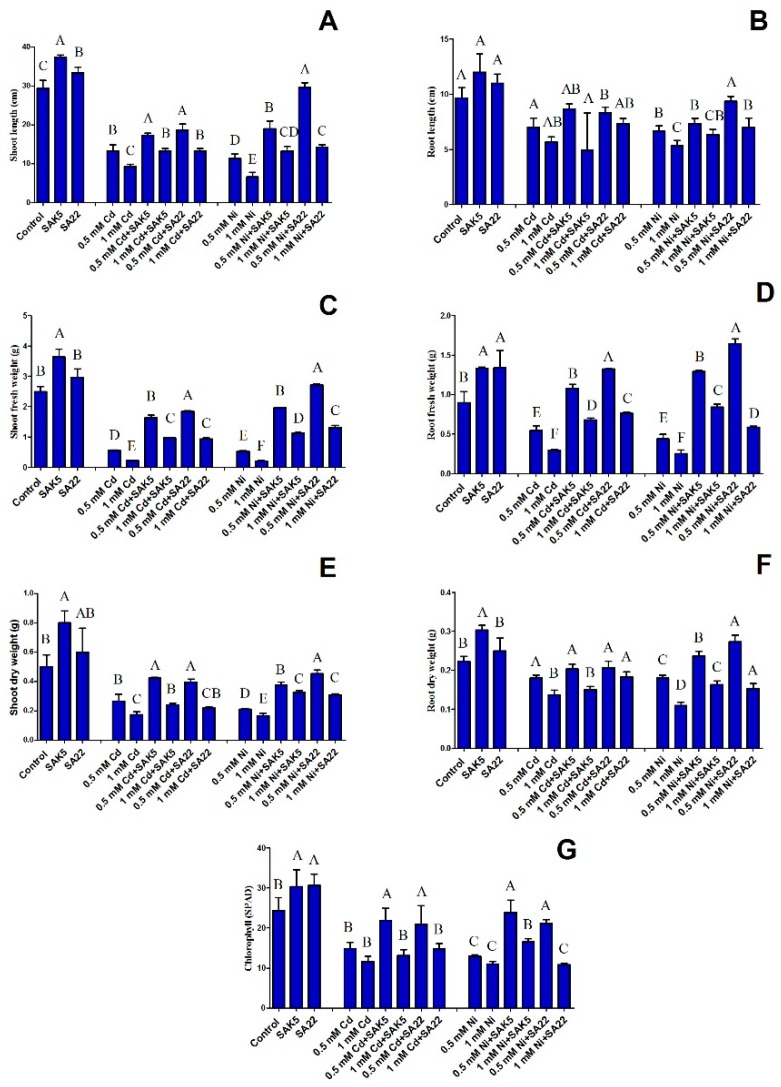
Effect of SAK5 and SA22 on the growth of rice seedlings. (**A**) Shoot length, (**B**) root length, (**C**) shoot fresh weight, (**D**) root fresh weight, (**E**) shoot dry weight, **(F**) root dry weight, and (**G**) chlorophyll content under control conditions, cadmium stress (0.5 mM and 1 mM Cd), and nickel stress (0.5 mM and 1 mM Ni). Control, untreated plants; SAK5, *Enterobacter ludwigii*-treated plants; SA22, *Exiguobacterium indicum*-treated plants; cadmium, 0.5 mM and 1 mM cadmium-treated plants; 0.5 mM and 1 mM Cd + SAK5, cadmium + SAK5-treated plants; 0.5 mM and 1 mM Cd + SA22, cadmium + SA22-treated plants; nickel, 0.5 mM and 1 mM nickel-treated plants; 0.5 mM and 1 mM Ni + SAK5, nickel + SAK5-treated plants; 0.5 mM and 1 mM Ni + SA22, nickel + SA22-treated plants. Data are means of three replicates along with standard error bars. Mean bars labeled with different letters are significantly different (*p* < 0.05), as evaluated by DMRT analysis.

**Figure 4 plants-08-00363-f004:**
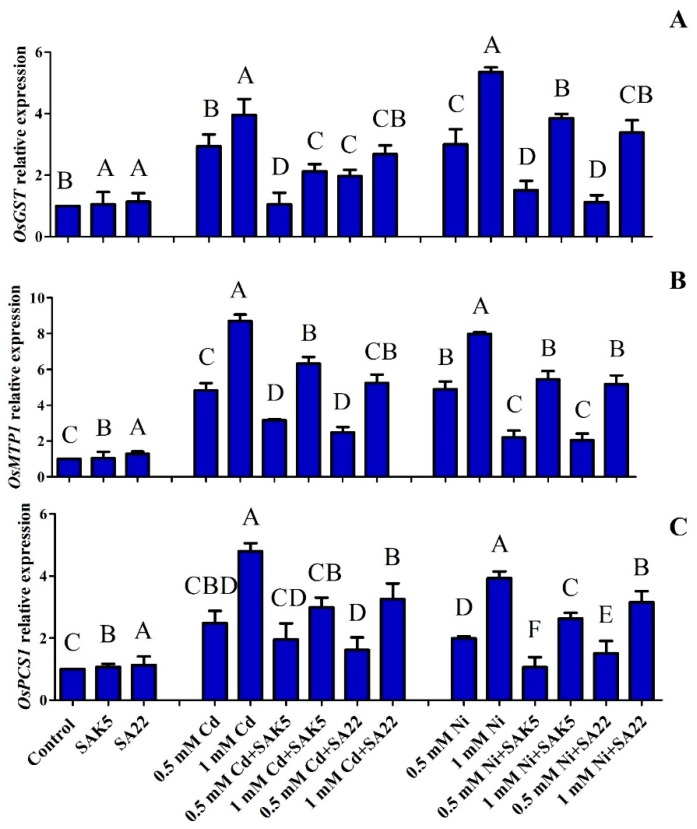
(**A**) *OsGST1* gene expression regulation following inoculation with SAK5 and SA22. (**B**) *OsMTP1* and (**C**) *OsPCS1* in rice plants under control conditions, cadmium stress (0.5 mM and 1 mM), and nickel stress (0.5 mM and 1 mM). Each value is the mean of three replicates. Data are means of three replicates along with standard error bars. Mean bars labeled with different letters are significantly different (*p* < 0.05), as evaluated by DMRT analysis.

**Figure 5 plants-08-00363-f005:**
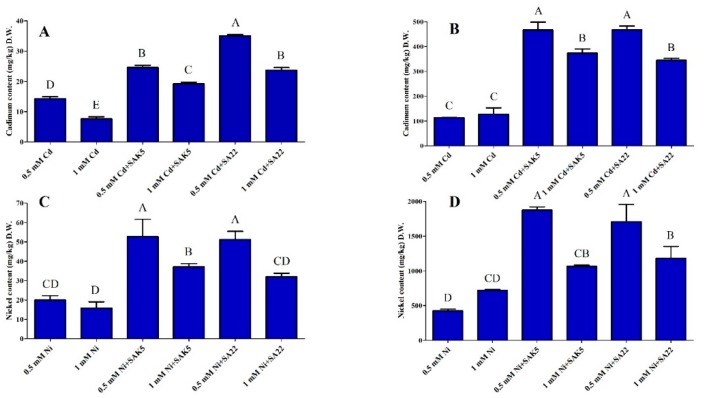
ICP analysis of the Cd and Ni content in the (**A,C**) shoots and (**B,D**) roots of rice plants under control conditions, cadmium stress (0.5 mM and 1 mM), and nickel stress (0.5 mM and 1 mM) following inoculation with SAK5 and SA22. Each value is the mean of three replicates. Data are means of three replicates along with standard error bars. Mean bars labeled with different letters are significantly different (*p* < 0.05), as evaluated by DMRT analysis.

**Figure 6 plants-08-00363-f006:**
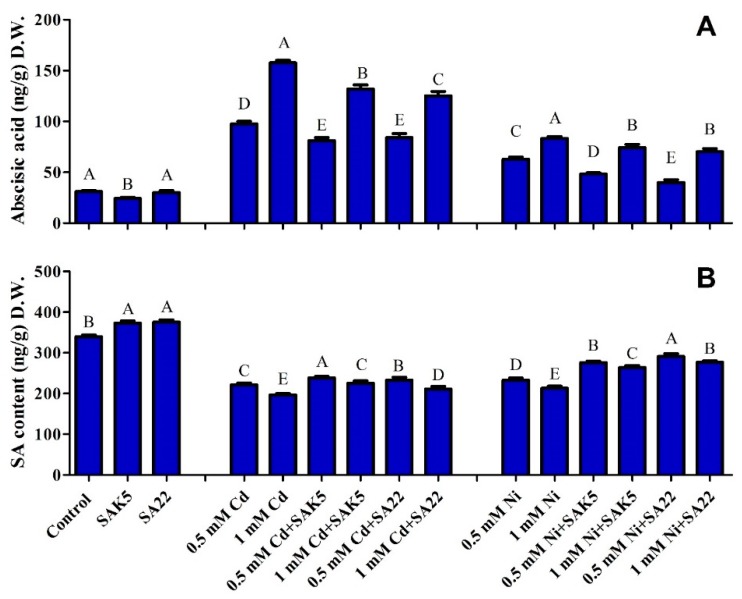
(**A**) Abscisic acid and (**B**) salicylic acid levels under cadmium stress (0.5 mM and 1 mM) and nickel stress (0.5 mM and 1 mM) in SAK5-inoculated, SA22-inoculated, and non-inoculated rice plants. Each value is the mean of three replicates. Data are means of three replicates along with standard error bars. Mean bars labeled with different letters are significantly different (*p* < 0.05), as evaluated by DMRT analysis.

**Figure 7 plants-08-00363-f007:**
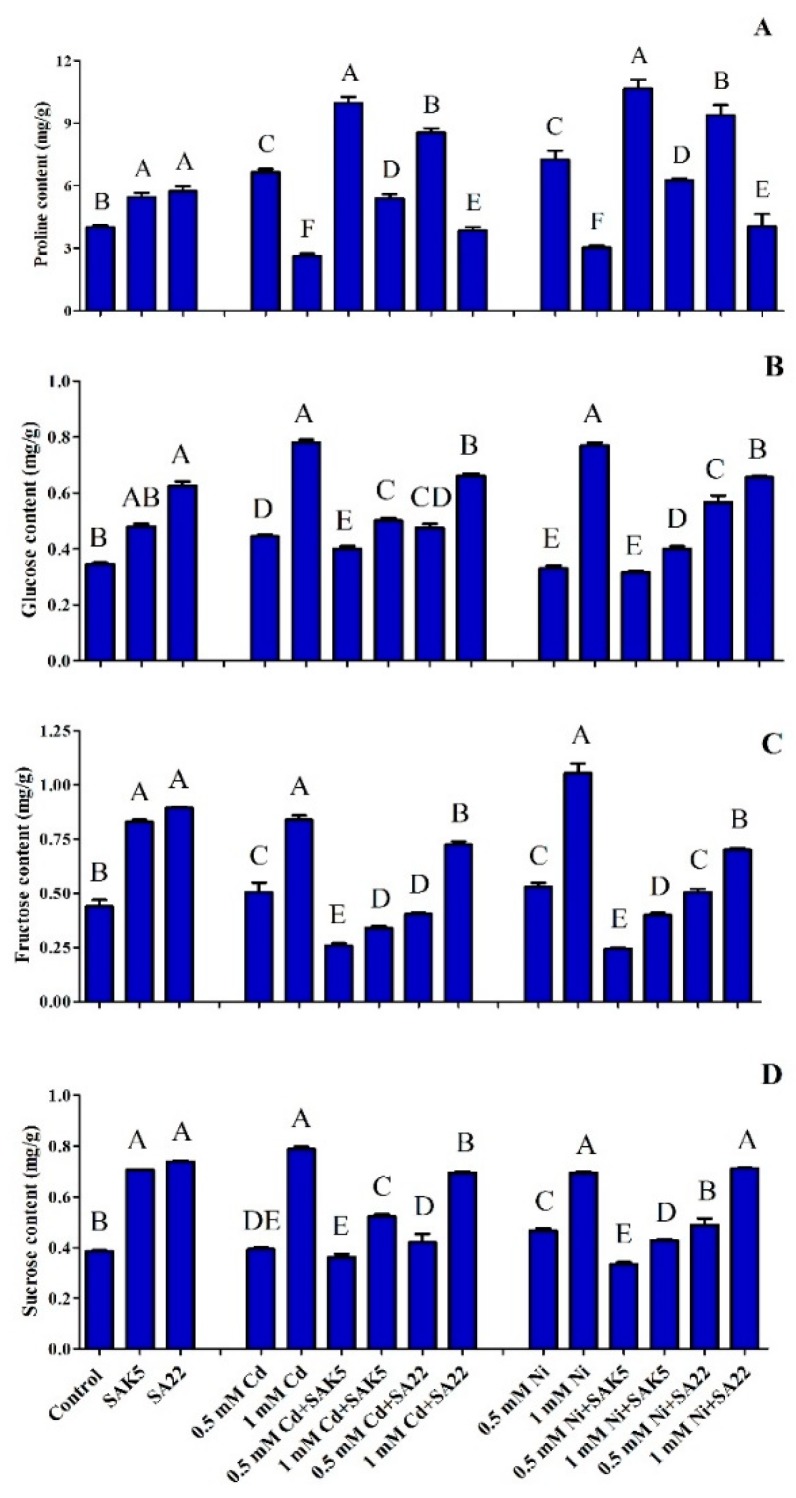
(**A**) Proline regulation, (**B**) total sugar content (**C**), glucose concentration, (**D**) fructose concentration, and (**E**) sucrose concentration under control conditions, cadmium stress (0.5 mM and 1 mM), and nickel stress (0.5 mM and 1 mM) in SAK5-inoculated, SA22-inoculated, and non-inoculated rice plants. Each value is the mean of three replicates. Data are means of three replicates along with standard error bars. Mean bars labeled with different letters are significantly different (*p* < 0.05), as evaluated by DMRT analysis.

**Table 1 plants-08-00363-t001:** List of genes, its primers and accession number.

Gene Name	Sequence (5′-3′)	Accession Number
OsGST-F	ATGGCGGCGGCGGAGAAGACGAA	XM_015764302
OsGST-R	TCAGGATGAAGGTGCATCTGGTTGG	
OsMTP1-F	ATGGACAGCCATAACTCAGCACCTCCCC	XM_015784797
OsMTP1-R	CTACTCGCGCTCAATCTGAATGGTTC	
OsPCS1-F	ATGCGATGCACATTCCCTTTGC	LC192427
OsPCS1-R	TTAGCATTGTTCCCAAGGTTGTGG

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
