# Peer review of "Metal Resistant Endophytic Bacteria Reduces Cadmium, Nickel Toxicity, and Enhances Expression of Metal Stress Related Genes with Improved Growth of Oryza Sativa, via Regulating Its Antioxidant Machinery and Endogenous Hormones"

_plants, 2019, doi:10.3390/plants8100363_

Round 1

Reviewer 1 Report

I would like to congratulate the authors for the descriptive work. This manuscript is an interesting analysis and characterizing of 2 isolated metal resistant endophytic bacteria and their growth promoting influence on Oryza sativa.

Generally, a control in the form of a non-metal resistant bacterium would be useful as a zero control to weight the results. Since without this control the question remains whether the shown effects are based on the metal resistance or is a general bacterial effect.

Another useful control would be a metal resistant non endophytic soil bacterium to distinguish the influence of in-planta and ex-planta metal accumulation.

Overall, the totality of data collected shows a direct impact of the metal-accumulating bacterial strains on plant growth and stress parameters. This will underpin a bridge in bioremediation between plants and bacterial endophytes.

The discussion should be shortened and focused and not largely repeat the results or be filled with many text passages from the literature.

Some minor point:

11 – “..concentrations of Cadmium (Cd) and Nickle (Ni).” 23/25 – “…while SA were found synergistic to ABA.” - If abbreviations such as Cd are written out, abbreviations such as SA and ABA should also be described.

119 –“..-IAA production using the method of…” - The same applies to IAA, starting in line 119. (see SA and ABA).

56 – “Additionally Cd causes cell injury due to generation of reactive oxygen species (ROS),…” - But not directly as redox inactive metal.

146/147 – “For calculation the concentration of IAA contents in the broths were calculated by comparing the peak areas of IAA with the known standard by using gas chromatography mass spectrometry (GC-MS) in selected ion monitoring mode (SIM).” - Related to what (cell numbers, dry weight,…)? See also figure 1 (D), µg/ml what?!

170/171 – “Before inoculation both the strains were grown in LB liquid medium until OD 600 and then spell down about 3-4 grams...” – What does it mean, until OD600?

196 - 2.7. Cd and Ni uptake analysis of ICP-MS - The method description is incomplete. Was an internal standard, external calibration, ... used? Were determination and detection limits determined using blanks?

Figure 1 (B) – y-axis –“ Cd and Ni quantification“ – What is the unit?

290 „The bacterial isolate such as

291 E. ludwigiiSAK5 produce the highest amount (2.7 ± 0.7 μg mL-1) – Per ml of what?

365/366 – “Different letters indicate significant difference different from each other as evaluated by DMRT analysis.” – This information is missing in legend of figure 1& 2 is missing. And the meaning of each letter is not explained!

372 - ”The capacity of accumulation of both the metals were increased at 0.5mM concentration in both the inoculated and non-inoculated…” - Compared to what?

502 – “…the uptake rate of SAK5 were higher (631 and 610mg/kg Cd and Ni respectively) than SA22 strain…” – Per kg of what?

531/532 –“Our results validated by previous report which confirmed thatglutathione mutant Arabidopsis showed hypersensitivity to both Cd and Cu stress [95, 96]. - This inference cannot be understood directly from the experiments carried out!

585 – “QTLs localization also identified OsMTP1 gene as a highly…” – QTLs? Abbreviations should be explained!

589/590 – “We isolated single phytochelatins (PCs) synthesis gene (OsPCS1) from rice plant.” - Context and meaning in this context unclear!

601- 603 – “In the present study it is predicted that in inoculated plants, bacteria enhanced GHS level which provide substrate for OsPCS1 and enhanced accumulation as well as tolerance of Cd and Ni.” - This statement is not comprehensible, since corresponding experiments on the GSH content in the plant (with and without inoculation) were not carried out.

Author Response

Reviewer 1

Comments and Suggestions for Authors

I would like to congratulate the authors for the descriptive work. This manuscript is an interesting analysis and characterizing of 2 isolated metal resistant endophytic bacteria and their growth promoting influence on Oryza sativa.

Generally, a control in the form of a non-metal resistant bacterium would be useful as a zero control to weight the results. Since without this control the question remains whether the shown effects are based on the metal resistance or is a general bacterial effect.

Reply:  Thank you very much for review and useful suggestions. Yes, you are right that a zero control would be useful to weight the results. As we stated in MS that we selected these bacteria both on the basis of plant growth promoting characteristics and heavy metal resistance. Our screening results revealed (not shown here) that these two strains have such plant growth promoting characteristics and heavy metal resistance. All these results suggested that the effect of these strains based on both plant growth promoting as well as heavy metal resistance.  

Another useful control would be a metal resistant non endophytic soil bacterium to distinguish the influence of in-planta and ex-planta metal accumulation.

Reply: Yes, you are right that another metal resistant bacterium could be used as a useful control in this study. However, in this study we only focused on endophytic bacteria which is useful for plant growth promotion as well.  We have isolated various bacteria from contaminated soil which showed activity against heavy metal stress but unable to promote plant growth as compared to control. We really like your suggestion related to different control and surely, we will include in future experiment related to bioremediation.

Overall, the totality of data collected shows a direct impact of the metal-accumulating bacterial strains on plant growth and stress parameters. This will underpin a bridge in bioremediation between plants and bacterial endophytes.

The discussion should be shortened and focused and not largely repeat the results or be filled with many text passages from the literature.

Reply:  Thank you for your suggestion, discussion is reduced to the possible level. 

Some minor point:

11 – “..concentrations of Cadmium (Cd) and Nickle (Ni).” 23/25 – “…while SA were found synergistic to ABA.” - If abbreviations such as Cd are written out, abbreviations such as SA and ABA should also be described.

Reply:  Corrected, now line number 32.

119 –“..-IAA production using the method of…” - The same applies to IAA, starting in line 119. (see SA and ABA).

 Reply:  Corrected, now line number 133.

56 – “Additionally Cd causes cell injury due to generation of reactive oxygen species (ROS),…” - But not directly as redox inactive metal.

 Reply:  Thank you for your comment, you are right, but actually here we mean that heavy metal generate reactive oxygen species which causes oxidative burst which is one of the mean reason for cell injury. Now line number 66.

146/147 – “For calculation the concentration of IAA contents in the broths were calculated by comparing the peak areas of IAA with the known standard by using gas chromatography mass spectrometry (GC-MS) in selected ion monitoring mode (SIM).” - Related to what (cell numbers, dry weight,…)? See also figure 1 (D), µg/ml what?!

Reply:  It is mention in methodology that we used bacterial culture filtrate, further informations we added in figure 1. Legend, now line number 938.

170/171 – “Before inoculation both the strains were grown in LB liquid medium until OD 600 and then spell down about 3-4 grams...” – What does it mean, until OD600?

Reply:  OD is the optical density, used for measurement of bacterial growth, corrected in text as well. Now line number 188.

196 - 2.7. Cd and Ni uptake analysis of ICP-MS - The method description is incomplete. Was an internal standard, external calibration, ... used? Were determination and detection limits determined using blanks?

Reply:  Detailed method added into MS. Now line number 215.

Figure 1 (B) – y-axis –“Cd and Ni quantification“ – What is the unit?

 Reply; Corrected

290 „The bacterial isolate such as

291 E. ludwigiiSAK5 produce the highest amount (2.7 ± 0.7 μg mL-1) – Per ml of what?

Reply:  It is mentioned in the methodology that we used bacterial culture filtrate for isolation of IAA, so here per ml mean ml of culture filtrate. Generally researchers using the same unit (ug/ml) for hormonal quantification line number 144. The information also added into MS, Now line number 291.

365/366 – “Different letters indicate significant difference different from each other as evaluated by DMRT analysis.” – This information is missing in legend of figure 1& 2 is missing. And the meaning of each letter is not explained!

 Reply:  Data are means of three replicates along with standard error bars. Mean bars labeled with different letters are significantly different (p < 0.05) as evaluated by DMRT analysis. Related information added into respective figure legends.    

372 - ”The capacity of accumulation of both the metals were increased at 0.5mM concentration in both the inoculated and non-inoculated…” - Compared to what?

 Reply:  Compared to 1.0mM concentration, information added to MS now line number 359.

502 – “…the uptake rate of SAK5 were higher (631 and 610mg/kg Cd and Ni respectively) than SA22 strain…” – Per kg of what?

 Reply:  Thank you for your comment, it was typo mistake, it is µg/ml of culture filtrate, now line number 465.

531/532 –“Our results validated by previous report which confirmed tha tglutathione mutant Arabidopsis showed hypersensitivity to both Cd and Cu stress [95, 96]. - This inference cannot be understood directly from the experiments carried out!

 Reply:  Actually this statement support our data because, we mentioned in our study that expression of OsGST gene is responsible for glutathione production which reduces heavy metal stress, similarly glutathione mutant Arabidopsis is sensitive to heavy metal stress. Now line number 486.

585 – “QTLs localization also identified OsMTP1 gene as a highly…” – QTLs? Abbreviations should be explained!

Reply:  Corrected, now line number 520.

589/590 – “We isolated single phytochelatins (PCs) synthesis gene (OsPCS1) from rice plant.” - Context and meaning in this context unclear!

 Reply:  Removed the statement, thank you.

601- 603 – “In the present study it is predicted that in inoculated plants, bacteria enhanced GHS level which provide substrate for OsPCS1 and enhanced accumulation as well as tolerance of Cd and Ni.” - This statement is not comprehensible, since corresponding experiments on the GSH content in the plant (with and without inoculation) were not carried out.

Reply:  Thank you for your valuable comment, firstly GHS was a typo mistake actually it mean glutathione (GSH) which is now change to glutathione throughout the MS, secondly on the basis of previous reports, we assumed that OsPCS1 uses glutathione as a substrate acting as a chelator of heavy metal. Previous report shows that endophytic bacteria increases glutathione (Fengshan, P. et al., 2016). However over results also shows that the selected strains produces glutathione (see Figure 1C). Now line number 535-536.

Fengshan, P.; Qian, M.; Qiong, W.; Sha, L.; Bao, C.; Kiran, Y. K.; Xiaoe, Y.; Ying, F., Endophytic bacterium Sphingomonas SaMR12 promotes cadmium accumulation by increasing glutathione biosynthesis in Sedum alfredii Hance. Chemosphere 2016, 154, 358-366.

Reviewer 2 Report

the authors present a huge work to describe the effects of plant growth promoting endophytic (PGPE) bacteria against 11 varying concentrations of Cadmium (Cd) and Nickle (Ni).

However, according to my opinion the results are not exhaustive. I suggest to introduce results on the rice production, on the rice quality and other parameters related to the crop.

The results description and presentation need to be improved.

Typos error are present.

Author Response

Reviewer 2

Comments and Suggestions for Authors

the authors present a huge work to describe the effects of plant growth promoting endophytic (PGPE) bacteria against 11 varying concentrations of Cadmium (Cd) and Nickle (Ni).

However, according to my opinion the results are not exhaustive. I suggest to introduce results on the rice production, on the rice quality and other parameters related to the crop.

The results description and presentation need to be improved.

Typos error are present.

Reply: Thank you very much for timely review and valuable suggestions. You are right that if we could include more information related rice production and rice quality it would greatly improve our manuscript. But actually our study is only related to plant growth and heavy metal accumulation so we only focus on that side.  Also it would be a huge and lengthy work if we try to include results related production and quality of rice. However in our new project we are focusing on rice quality and quantity. Furthermore, we have improved the results section as well according to the reviewer 1 and reviewer 3 suggestions. We hope this time it will according to reviewer suggestions.

Reviewer 3 Report

This author isolated endophytic bacteria from various plants. And then they screened six endophytic bacteria, which are enhanced rice growth. Furthermore, two endophytic bacteria (SAK5 and SA22) out of six showed Cd and Ni resistance phenotype. They performed further experiment using SAK5 and SA22 inoculated rice plants. They compared, growth, metal concentration, gene expression, hormone and sugar content. Taken together inoculation of SAK5 or SA22 was enhanced Cd and Ni tolerance in rice. I think this phenomenon is interesting and might be important information for agricultural technology in future. But I claim to improve some experiment.

Fig1A. Author showed growth of SAK5 and SA22 in Cd and Ni plate. But there is no control. It is difficult to judge SAK5 and SA22 is high Ni and Cd tolerance from this data. On the other hand Author have other bacteria data. Page 7 line 243 “The results revealed that among six endophytic bacterial strains, only SAK5 and SA22 were highly tolerant to Cd and Ni and were showed a normal growth in Cd and Ni supplemented PAD media, while the other four endophytes were sensitive to Cd and Ni, as their growth were inhibited by these metals.” I think you should show the growth data to compare SAK5, SA22 and other bacteria with or without Ni and Cd.

Furthermore SAK5 and SA22 tolerance is specific for Cd and Ni ? How about resistance of other heavy metals ?

Fig1D Author compared IAA quantification in SAK5 and SA22. I can’t understand why did you determine IAA contents. Could you explain the purpose of this experiment.

Fig 2 Author compared rice growth inoculated bacteria with or without Ni and Cd solution. I think this is most important experiment. But your picture has some problems.

Why some solutions have green color (this is algae ? ) ? I think there is some other bacteria (not only SAK5 or SA22) or algae effect on rice growth.

Author exposed to two different metal concentration (0.5 and 1 mM). But plants almost died in at least 0.5 mM condition. Author should show more mild condition (0.1 or 0.05 mM etc).

And you should add scale bar in Fig 2.

Fig 5 Author determined Ni and Cd concentration. The inoculated plant increased Ni and Cd concentration in both root and shoot. This data means uptake ability and accumulation capacity of Ni and Cd is enhanced by inoculation. In plant Ni and Cd acquisition is compete other essential metal. Could you show other metal (Cu, Zn, Fe, Mn etc) concentration in root and shoot.

The title of this paper is “Metal Resistant Endophytic Bacteria Reduces Cadmium, Nickel Toxicity and Expression of Metal Transporter Genes with Improved Growth of Oryza sativa, via Regulating its Antioxidant Machinery and Endogenous Hormones”

But in this paper showed OsGST, OsMTP1 and OsPCS1 expression. Could you consider title again. Because OsGST, OsPCS1 are not transporter.

Page 4 Line 157; Rice seeds obtained from Plant Molecular Breeding lab Kyungpook
You should be write down which cultivar used.

Page 5 Line 164; After successful growth the seedling were transplanted to hydroponic medium (Hoagland solution pH 5.8)
Author used Hoagland solution for hydroponic condition in all experiment. I think Hoagland solution is for upland plant (Arabidopsis or barley etc.) Usually we used Kimura B or 1/2 Kimura B solution in rice hydroponic culture. Nitrogen souse and other element concentration is different between Hoagland and Kimura B solution.

You should insert a space between number and unit symbol. (ex. Page 5 line 171; 40ml distilled water should be 40 ml distilled water.)

Page 5 Line 180; Chlorophyll contents were measured in three replicates using chlorophyll meter (SPAD-502 Minolta, Japan). Which leaf did you measure?

Page 7 Line 238; Experiments were performed in triplicate and the values obtained are presented as the means ± standard deviation (SD). This is all experiments? Figure legend 5,6,7 showed Error bar represent standard error. SD and SE is different calculation.

Page 7 Fig1A, Fig1C; Characters in the figure should be lowercase. (Fig 1A (a), (b), (c)….)

Page 7 Fig1B; There is no unit. Could you add unit.

Page 12 Line 392 Could you confirm font size.

Page 13 Fig 6 Abscisic acid (ng/g DW) Could you change to ABA content (ng/g) D.W.

Page 14 Line 456 Could you confirm font size.

Author Response

Reviewer 3

Reviewer #3 comments are highlighted with yellow color.

Comments and Suggestions for Authors

This author isolated endophytic bacteria from various plants. And then they screened six endophytic bacteria, which are enhanced rice growth. Furthermore, two endophytic bacteria (SAK5 and SA22) out of six showed Cd and Ni resistance phenotype. They performed further experiment using SAK5 and SA22 inoculated rice plants. They compared, growth, metal concentration, gene expression, hormone and sugar content. Taken together inoculation of SAK5 or SA22 was enhanced Cd and Ni tolerance in rice. I think this phenomenon is interesting and might be important information for agricultural technology in future. But I claim to improve some experiment.

Fig1A. Author showed growth of SAK5 and SA22 in Cd and Ni plate. But there is no control. It is difficult to judge SAK5 and SA22 is high Ni and Cd tolerance from this data. On the other hand Author have other bacteria data. Page 7 line 243 “The results revealed that among six endophytic bacterial strains, only SAK5 and SA22 were highly tolerant to Cd and Ni and were showed a normal growth in Cd and Ni supplemented PAD media, while the other four endophytes were sensitive to Cd and Ni, as their growth were inhibited by these metals.” I think you should show the growth data to compare SAK5, SA22 and other bacteria with or without Ni and Cd.

Reply: Thank you for your review and suggestions. We have incorporated data and supplementary results according to your suggestions. Now control (0 mM of Cd and Ni) for the SAK5 and SA22 is presented in Supplementary (Fig. S3) data, and screening data of the remaining four bacteria are presented in figure S4. Also mention in the text with 268 and 270 line no respectively.

Furthermore SAK5 and SA22 tolerance is specific for Cd and Ni? How about resistance of other heavy metals?

Reply: You are right both the strains may have tolerance to other metals as as well. In this study we only checked Cd and Ni because of the study site contamination and project requirement.  In future we are planning to use these microbes against different heavy metal stress as well as  salinity and drought stresses.

 Fig1D Author compared IAA quantification in SAK5 and SA22. I can’t understand why did you determine IAA contents. Could you explain the purpose of this experiment.

Reply: As IAA is a common plant growth promoter and consider as a common trait of plant growth promoting bacteria. We checked IAA production because the aim of our study is to evaluate the plant growth promoting activity of both the strains along with heavy metal tolerance.  

Fig 2 Author compared rice growth inoculated bacteria with or without Ni and Cd solution. I think this is most important experiment. But your picture has some problems.

Why some solutions have green color (this is algae ? ) ? I think there is some other bacteria (not only SAK5 or SA22) or algae effect on rice growth.

Reply: Yes, you are right this is one of the important experiments in this study. We have replaced the old figure with new figure and there is no bacterial contamination in these pots. However, we could not take good quality picture form these pots and took it from different angles, distance and light. In another study we have repeated the same experiments in pots having soil with rice as well as with soybean and tomato and got same results (but not part of this study and projects). Therefore, we are sure that the plant growth promoting effect was due to these bacterial strains not contamination.

Author exposed to two different metal concentration (0.5 and 1 mM). But plants almost died in at least 0.5 mM condition. Author should show more mild condition (0.1 or 0.05 mM etc).

Reply: Thank you for your suggestion, along with 0.5- and 1.0-mM concentrations, we also checked 0.3 mM concentration but we couldn’t find significant difference between control and 0.3 mM treated plants, that’s why we ignored that concentration. 

And you should add scale bar in Fig 2.

Reply: We have added scale bar to Fig 2 according to your suggestion and change the figure as well.

Fig 5 Author determined Ni and Cd concentration. The inoculated plant increased Ni and Cd concentration in both root and shoot. This data means uptake ability and accumulation capacity of Ni and Cd is enhanced by inoculation. In plant Ni and Cd acquisition is compete other essential metal. Could you show other metal (Cu, Zn, Fe, Mn etc) concentration in root and shoot.

Reply:  You are right that Ni and Cd accumulation compete with other essential metals and their concentration might be increased or decreased due to Cd and Ni accumulation, but actually our study is concern to Ni and Cd accumulation which we applied exogenously that’s why we checked only these two metals concentrations. Unfortunately, we don’t have dry plant sample to analyze for other metal concentrations.

The title of this paper is “Metal Resistant Endophytic Bacteria Reduces Cadmium, Nickel Toxicity and Expression of Metal Transporter Genes with Improved Growth of Oryza sativa, via Regulating its Antioxidant Machinery and Endogenous Hormones”

But in this paper showed OsGST, OsMTP1 and OsPCS1 expression. Could you consider title again. Because OsGST, OsPCS1 are not transporter.

Reply:  Thank you for your important suggestion, I hope this title will be more suitable for the MS. ’’Metal Resistant Endophytic Bacteria Reduces Cadmium, Nickel Toxicity and Enhances Expression of Metal stress related Genes with Improved Growth of Oryza sativa, via Regulating its Antioxidant Machinery and Endogenous Hormones’’.

Page 4 Line 157; Rice seeds obtained from Plant Molecular Breeding lab Kyungpook

You should be write down which cultivar used.

Reply: Ilmi cultivar was used in this study, added in text, now line number 174.

Page 5 Line 164; After successful growth the seedling were transplanted to hydroponic medium (Hoagland solution pH 5.8)

Author used Hoagland solution for hydroponic condition in all experiment. I think Hoagland solution is for upland plant (Arabidopsis or barley etc.) Usually we used Kimura B or 1/2 Kimura B solution in rice hydroponic culture. Nitrogen souse and other element concentration is different between Hoagland and Kimura B solution.

Reply: Thank you for sharing your information, actually we have no experience with Kimura B and mostly we using Hoagland solution for rice and frequently researcher using it for rice such as (John Milton Lima, et al. 2015). Now line number 183. https://doi.org/10.1093/aobpla/plv023

You should insert a space between number and unit symbol. (ex. Page 5 line 171; 40ml distilled water should be 40 ml distilled water.)

Reply: Corrected, now line number 189.

Page 5 Line 180; Chlorophyll contents were measured in three replicates using chlorophyll meter (SPAD-502 Minolta, Japan). Which leaf did you measure?

Reply: we selected third leaf of the plant for chlorophyll measurement, now line number 195.

Page 7 Line 238; Experiments were performed in triplicate and the values obtained are presented as the means ± standard deviation (SD). This is all experiments? Figure legend 5,6,7 showed Error bar represent standard error. SD and SE is different calculation.

Reply: Thank you for your guidance, data are means of three replicates along with standard error bars, added to respective figure legends. It is also mentioned in methodology (2.11. Statistical analysis). Now line number 260.

Page 7 Fig1A, Fig1C; Characters in the figure should be lowercase. (Fig 1A (a), (b), (c)….)

Reply: Corrected

Page 7 Fig1B; There is no unit. Could you add unit.

Reply: Corrected

Page 12 Line 392 Could you confirm font size.

Reply: Corrected

Page 13 Fig 6 Abscisic acid (ng/g DW) Could you change to ABA content (ng/g) D.W.

Reply: Corrected

Page 14 Line 456 Could you confirm font size.

Reply: Corrected

Round 2

Reviewer 2 Report

The author improve the paper after the revision.